# Antenatal care booked rural residence women have home delivery during the era of COVID-19 pandemic in Gidan District, Ethiopia

**Desale Bihonegn Asmamaw** [1] *, **Yohannes Ayanaw Habitu**[1], **Eskedar Getie Mekonnen**[1], **Wubshet Debebe Negash**[2]

**1** Department of Reproductive Health, Institute of Public Health, College of Medicine and Health Sciences, University of Gondar, Gondar, Ethiopia, **2** Department of Health Systems and Policy, Institute of Public Health, College of Medicine and Health Sciences, University of Gondar, Gondar, Ethiopia

* desalebihonegn1988@gmail.com

**Data Availability Statement:** All relevant data are within the paper and Supporting Information file.

## Abstract

### Background

World Health Organization (WHO) recommends that every pregnant woman receive quality care throughout pregnancy, childbirth, and the postnatal period. It is estimated that institutional delivery could reduce 16% to 33% of maternal deaths. Despite the importance of giving birth at a health institution, in Ethiopia, according to the Ethiopian Demographic Health Survey report, nearly half of the ANC-booked mothers gave birth at home. Therefore, this study aimed to determine the prevalence and associated factors of home delivery among antenatal care-booked women in their last pregnancy during the era of COVID-19.

### Methods

A community-based cross-sectional study was conducted from March 30 to April 29, 2021. A simple random technique was employed to select 770 participants among women booked for antenatal care. Interviewer-administered questionnaires were used to collect the data. A binary logistic regression model was fitted. Adjusted odds ratios with its respective 95% confidence interval were used to declare the associated factors.

### Results

The prevalence of home delivery was 28.8% (95% CI: 25.7, 32.2). Rural residence (AOR = 2.02, 95% CI: 1.23, 3.34), unmarried women (AOR = 11.16, 95% CI: 4.18, 29.79), husband education (AOR = 2.60, 95% CI: 1.72, 3.91), not being involved in the women's development army (AOR = 1.64, 95% CI: 1.01, 2.65), and fear of COVID-19 infection (AOR = 3.86, 95% CI: 2.31, 6.44) were significantly associated factors of home delivery.

### Conclusion

Even though the government tried to lower the rate of home delivery by accessing health institutions in remote areas, implementing a women's development army, and introducing maternal waiting home utilization, nearly one in every three pregnant women gave birth at

**Funding:** The author(s) received no specific funding for this work.

**Competing interests:** The authors have declared that no competing interests exist.

**Abbreviations:** ANC, Antenatal Care; AOR, Adjusted Odds Rati; CI, Confidence Intervals; COR, Crude Odds Ratio; COVID-19, Coronavirus disease; EDHS, Ethiopian Demographic, and Health Survey; LICs, Low-Income Countries; OR, Odds Ratio; PNC, Postnatal Care; WDA, Women Development Army; SDGs, Sustainable Development Goals; WHO, World Health Organization.

home among ANC booked women in their last pregnancy. Thus, improving the husband's educational status, providing information related to health institution delivery benefits during antenatal care, and strengthening the implementation of the women's development army, particularly among rural and unmarried women, would decrease home childbirth practices.

## Background

Maternal mortality is one of the most common public health problems in low and middle-income countries [1]. Reduction of maternal mortality remains a priority agenda under Goal 3 in the United Nations Sustainable Development Goals (SDGs) through 2030. This calls for the ambition of a maternal mortality ratio reduction of less than 70 per 100,000 live births between 2016 and 2030 [2]. Worldwide, approximately 295,000 women died during and after pregnancy and childbirth. The majority (94%) of deaths occurred in developing countries, including Ethiopia. Sub-Saharan Africa (SSA) and Southern Asia accounted for approximately 86% of all maternal deaths worldwide. Of this, two-thirds of the deaths were from SSA [3,4]. The maternal mortality ratio (MMR) in low-income countries is 462 deaths per 100,000 live births [3]. Ethiopia is among the countries in which high maternal mortality has occurred. The MMR in Ethiopia is 412 deaths per 100,000 live births [3].

The continuum of maternal health care is crucial to saving the mother and the child. In addition, they are indicators of progress toward the achievement of sustainable development goals [5,6]. The World Health Organization endorsed the need for skilled care for all women during their labor and immediately afterward [7]. ANC follow-up has increased in many parts of the world. However, only 46% of developing countries and 26% of Ethiopia benefited from skilled personnel attendance. Attendance by skilled personnel is advocated as the single most important factor in preventing maternal mortality and morbidity [6,8].

In SSA, only 17.7% of women chose to deliver at the health institution [9]. If pregnant women give birth at home, it increases the risk of infection and postpartum hemorrhage [10]. As a result, maternal deaths from hemorrhage, prolonged or obstructed labor, ruptured uterus, severe pre-eclampsia and eclampsia, sepsis, and abortion complications continue to be a problem [11]. The aforementioned causes can be detected and managed early during ANC and the intrapartum period by existing and well-known medical interventions [12]. Most home deliveries, mainly those of unattended births, are not only a problem for the mother, but they also result in perinatal and neonatal morbidity and mortality. In the SSA, home births were found to have a 21% higher rate of perinatal mortality than facility births [13].

Home delivery in itself is not a cause of maternal and child deaths, it becomes a risk factor when home deliveries occur without a skilled attendant due to the possibility of infection or a lack of equipment in case of complications [14]. Attending an ANC visit does not mean that deliveries of pregnant women will take place at a health institution. Despite 62% of women attending ANC, three-fourths of women nationwide gave birth at home [8]. Another study from Ethiopia revealed that many women unpredictably did not give birth in health institutions despite antenatal care [6].

Because of the COVID-19 infection, countries are facing significant challenges to maintain high-quality, essential maternal and newborn health services [15]. Mothers with newborns and pregnant women may experience difficulty accessing healthcare services due to disruptions in transportation and lockdowns or fear of COVID-19 infection. Thus, pregnancy-related and newborn healthcare coverage declined by 10% due to the pandemic. This would result in 28,000 maternal deaths [15,16].

There are variety of literature reporting factors that affect the use of ANC services and the place of delivery [17–24], but there is no adequate literature explaining why ANC-booked women prefer home deliveries. This leaves maternal morbidity and mortality a public health problem. Therefore, this study aimed to determine the prevalence and associated factors of home delivery among women who had antenatal care booked in their last pregnancy during the era of COVID-19. By analyzing this study, policymakers will be able to find the appropriate strategy for improving maternal and child health in the area.

## Methods

### Study design and setting

A community-based cross-sectional study design was conducted from March 30 to April 29, 2021, in the Gidan district, North Wollo zone, Northeastern Ethiopia. The district is located 595 Kilometers (km) away from Addis Ababa, the capital city of Ethiopia, and it has two urban and 21 rural kebeles (the lowest administrative unit). It has an estimated 148,058 population based on the population projection from the 2007 census through 2020, of which 74,461 and 29,649 are females and reproductive age women, respectively. The district has six health centers and 23 health posts that provide routine health services for the catchment population [25]. The study was conducted from March 30 to April 29, 2021.

### Study population, sample size determination, and sampling procedures

The study population consisted of all ANC-booked mothers who gave birth within the last year and resided in the selected kebeles of the district. The required sample size for home delivery was calculated using the single population proportion formula, considering the following statistical assumptions: margin of error 5% (0.05), Z-value 1.96 corresponding to 95% confidence level, design effect 2, and the prevalence of home delivery was 35.2% [18], accordingly the sample size was computed as:

$$n = \frac{(Z\alpha 1/2)^2 \, P \, (1-P)}{d2}$$

$$n = \frac{(1.96)^2 \, (0.352) \, (1-0.352)}{(0.05)2} = 350$$

n = 350

After adding a 10% non-response rate and multiplying it by the design effect of 2, the total sample size of this study was 770 mothers. A simple random sampling technique was used to select the participants. From 23 kebeles, seven kebeles were selected by a simple random sampling technique using the lottery method. The list of ANC-booked mothers who gave birth in the last year was identified by the health center and health extension workers. The sample size was proportionally allocated to each selected kebele, considering the number of women. Mothers in the sampled kebeles were selected by using a simple random sampling technique (Open Epi Random Program version 3) (Fig 1). Regardless of the outcome, limiting the participants to only 1 year and after one month of giving birth was done to minimize potential recall bias and make the mother stabilized and comfortable. If the respondents were not available at home during the time of data collection, interviewers revisited the households three times, and when the interviewers failed to find the eligible respondent after three visits, the next household was included.

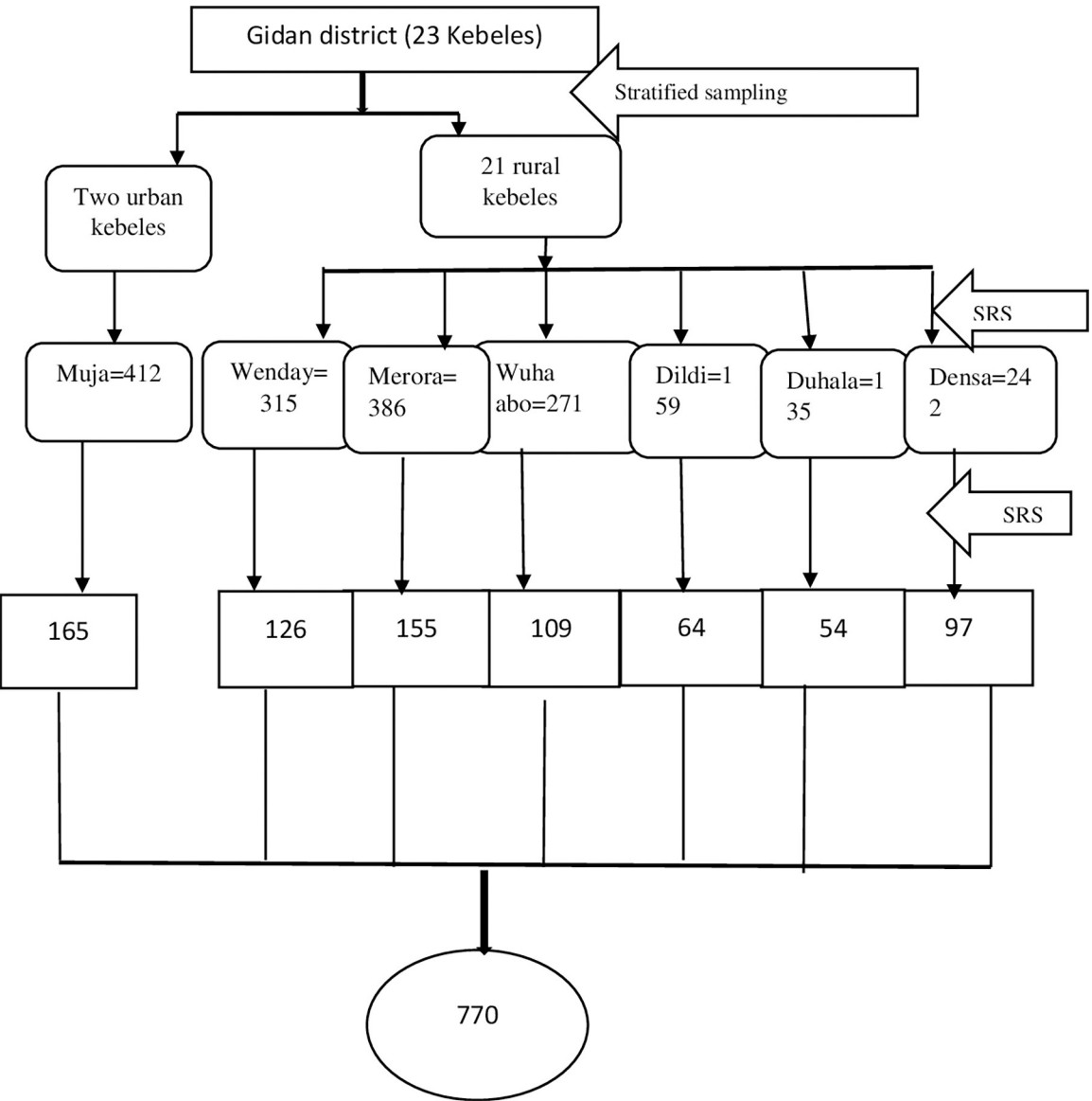

**Fig 1. Schematic presentation of the sampling procedure.**

## Study variables

The outcome variable was home delivery. when a woman gave birth at her home or others' homes (neighbors, relatives, or family) or when a birth took place outside of health institutions. It was dichotomized and coded as "0" and "1," representing those who had delivered at health institutions and home, respectively [26]. The independent variables were socio-demographic (age, educational status of women and husbands, occupation, marital status, residence), obstetrics-related characteristics (parity, pregnancy status, bad obstetric history), fear of COVID-19 infection, and other characteristics like membership in the Women's Development Army (WDA). Fear of COVID-19 infection means fear of oneself or family members being infected with COVID-19 infection during pregnancy when visiting health institutions [27].

## Data quality assurance

Data collectors and supervisors were trained for two days, focusing on how to ask and fill out the questionnaires, the selection criteria for women, and how to approach the respondents. Before starting the actual data collection, the data collectors practiced in the field, and the questionnaires were pretested on 39 study participants (5%) in the Gubalafto district. Findings and experiences from the pretest were utilized in modifying the data collection tool. The data collectors and the supervisors assessed the clarity and completeness of the filled-out questionnaires. The whole data collection process was closely supervised by the principal investigator (PI).

## Data collection tools and procedures

Structured interviewer-administered questionnaires developed from reviewing different related literature were used for data collection [18,28–33]. First, the questionnaires were developed in English. Then it was translated into Amharic (the local language) and retranslated back into the English language to check its consistency by language experts. The questionnaire was separated into different sections such as socio-demographic characteristics, obstetric-related characteristics of the participants, and other characteristics like fear of COVID-19 infection, lack of transport, and membership in WDA. Data collectors approached the women by introducing themselves and interviewing the selected respondent after informed consent had been obtained. Eight BSc midwifery or nursing data collectors and two supervisors in the same field with experience in research and fieldwork coordination participated in the data collection process.

## Data processing and analysis

The collected data were checked for completeness before being entered into Epi-data version 4.6 statistical software. Then it was exported to STATA version 14 statistical software for cleaning, coding, and analysis. Descriptive statistics were described using frequencies, percentages, mean, and standard deviation, which was further presented using tables, figures, and text. Normality tests such as kurtosis and skewness were employed to see the normal distribution of the variables and to identify which summary measures were appropriate to use. Before identifying the significant factors, multicollinearity was tested using the variance inflation factor (VIF), and we have a VIF of less than five for each independent variable with a mean VIF of 1.56, indicating there is no significant multicollinearity between independent variables [34]. According to the Hosmer-Lemeshow test, the model was adequate with a p-value of 0.26. The case-to-variable ratio showed 12.8 to 1, which indicates a ratio above the required 5 to 1.

A binary logistic regression analysis was carried out to identify factors associated with home delivery. Those variables with a P-value $\leq$ 0.25 from the bivariable analysis were entered into a multivariable logistic regression model to control the possible effects of confounder/s. The odds ratio with 95% confidence intervals was computed to see if there was an association between home delivery and associated factors. A P-value of 0.05 was used to declare a statistical association.

## Ethical considerations

Ethical clearance was obtained from the Institutional Review Board (IRB) of the University of Gondar (Ref. No: IPH/142/2013). Similarly, a supportive letter was taken from the district administrative office to be given to the selected kebeles. Verbal informed consent was obtained from each participant. Participants were also informed that participation was voluntary and

that they had the right to withdraw from the study at any time they wanted. All data obtained from participants was kept confidential and used only for this study. The study was conducted according to the Helsinki Declaration.

## Results

### Socio-demographic characteristics of the respondents

About 760 women participated, giving a response rate of 98.7%. The mean age of the study participants was 27 (SD±5.3) years, and 306 (40.3%) of them fell within the age category of 26–30 years. Three-fourths (75.4%) of the respondents were rural dwellers, and almost all (99.6%) participants were orthodox Christian followers (Table 1).

### Obstetrics and other related factors

Among the study participants, 612 (80.5%) and 650 (85.5%) of the mothers were multiparous and had an ANC visit for their index child, respectively. One hundred and fifty-six (20.5%) of the participants were involved in the Women's Development Army (WDA). About 532 (70%) of the mothers feared COVID-19 infection (Table 2).

### Place of delivery

The prevalence of home delivery in Gidan district was 28.8% (95% CI: 25.7%, 32.2%).

### Factors associated with home delivery

In bivariable logistic regression, rural residence, unmarried women, husband education, not being involved in WDA, and fear of COVID-19 infection were statistically significant factors for home delivery among ANC-booked women. Accordingly, the odds of home delivery were 2.02 times higher among rural residents (AOR = 2.02, 95% CI: 1.23, 3.34) compared to urban residents. Similarly, the odds of women having a home delivery were 11.16 (AOR = 11.16, 95% CI: 4.18, 29.79) times higher among unmarried women compared to married women.

Pregnant women who had no formally educated husbands had 2.60 times (AOR = 2.60, 95% CI: 1.72, 3.91) higher odds of home delivery compared to their counterparts. The likelihood of home delivery was 1.64 times higher (AOR = 1.64; 95% CI: 1.01, 2.65) among respondents who were not involved in WDA as compared to their counterparts. Home delivery was 3.86 times higher among women who feared COVID-19 infection (AOR = 3.86, 95% CI: 2.31, 6.44) than their counterparts (Table 3).

## Discussion

This study is a community-based study in randomly selected kebeles that includes both urban and rural settings. During pregnancy, childbirth, and the immediate postpartum period, the continuum of maternal health care is crucial. Even though health institutions are relatively accessible in remote areas, a significant number of pregnant women still give birth at home, contrary to the plan of health institution delivery. The purpose of this study was to determine the prevalence and associated factors of home delivery among ANC-booked mothers in their last pregnancy during the era of COVID-19 in Gidan District, Northeastern, Ethiopia.

The prevalence of women who gave birth at home among ANC-booked women was 28.8% (95%CI: 25.7–32.2). This finding is higher than studies conducted in Southern Ethiopia 22.8% [35] and in Bench Maji zone, Ethiopia 21.7% [36]. This difference might be due to the fear of COVID-19 transmissions and interruptions of services [37,38]. In addition, possibly due to internal conflicts (war) in the area, war can result in the distraction of infrastructure like roads,

**Table 1. Socio-demographic characteristics of the respondents in Gidan District, Northeastern Ethiopia, 2021 (N = 760).**

| Variables | Frequency(n) | Percentage (%) |
|---|---|---|
| **Age of the mother in years** | | |
| <20 | 74 | 9.7 |
| 20–25 | 197 | 25.9 |
| 26–30 | 306 | 40.3 |
| >30 | 183 | 24.1 |
| **Religion** | | |
| Orthodox | 757 | 99.6 |
| Muslim | 3 | 0.4 |
| **Educational status of the mother** | | |
| No formal education | 346 | 45.5 |
| Primary school | 302 | 39.7 |
| Secondary school and above | 112 | 14.8 |
| **Occupation of the mother** | | |
| Housewife | 645 | 84.9 |
| Government employed | 83 | 10.9 |
| Self-employed | 24 | 3.1 |
| Other * | 8 | 1.1 |
| **Current marital status** | | |
| Married | 671 | 88.3 |
| Single | 63 | 8.3 |
| Widowed | 18 | 2.4 |
| Other** | 8 | 1 |
| **Educational status of the husband** | | |
| No formal education | 361 | 53.8 |
| Primary school | 202 | 30.1 |
| Secondary school and above | 108 | 16.1 |
| **Occupation of the husband** | | |
| Farmer | 541 | 80.6 |
| Government employed | 51 | 7.6 |
| Self-employed | 61 | 9.1 |
| Other*** | 18 | 2.7 |
| **Residence** | | |
| Rural | 573 | 75.4 |
| Urban | 187 | 24.6 |

*student

**divorced, separated

***daily labor, soldier.

and most of the participants in the study areas are far from the health institutions, so it takes a long time to reach them by walking without an ambulance, and it is difficult to go to a health institution during the war. Scholars indicated that low maternal healthcare service utilization was higher in conflict areas than in non-conflict affected areas [39]. This seems to be an important determinant, which may preclude women from accessing delivery services.

On the other hand, the result of this study is lower than studies from the 2016 Ethiopian Health Survey report (EDHS), where 67.2% of the mothers delivered at home [22], and another Ethiopian pooled prevalence of 66.7% [40]. The possible justification for the difference

**Table 2. Obstetrics and other related factors of the respondent in Gidan District, Northeastern Ethiopia, 2021 (N = 760).**

| Variables | Frequency (n) | Percentage (%) |
|---|---|---|
| **Parity** | | |
| **Primipara** | **148** | **19.5** |
| **Multipara** | **612** | **80.5** |
| **Antenatal care** | | |
| Yes | 650 | 85.5 |
| No | 110 | 14.5 |
| **Pregnancy status** | | |
| Planned | 703 | 92.5 |
| Unplanned | 57 | 7.5 |
| **Membership for WDA** | | |
| Yes | 156 | 20.5 |
| No | 604 | 79.5 |
| **Bad obstetrics history** | | |
| Yes | 102 | 13.42 |
| No | 658 | 86.58 |
| **Interruption and diversion of maternal health care service** | | |
| Yes | 585 | 76.9 |
| No | 175 | 23.1 |
| **Fear of COVID-19** | | |
| Yes | 532 | 70 |
| No | 228 | 30 |
| **Lack of transport access** | | |
| Yes | 103 | 13.6 |
| No | 657 | 86.4 |

**Abbreviations**; Women development army; COVID-19, coronavirus diseases.

might be differences in the study area and source population. The current study included only women who had ANC follow-up, while the previous studies included women who did not have ANC follow-up. In addition to this, the decrease in home delivery in the current study could be because the government and other supporting organizations are working strongly to increase institutional delivery service utilization [17,21].

In addition, the strengthening of the health extension program and the implementation of WDA facilitate maternal health care services, including institutional delivery [41]. The establishment of maternal waiting homes also played a greater role (80% of the reduction of maternal deaths) in improving the current reduction in home delivery [42]. This implied that strengthening the above activities would result in a better outcome.

Women from rural dwellers had 2.02 times more odds of delivering at home than women from urban dwellers; this finding is consistent with studies conducted in Ethiopia [31], Uganda [43], and Nigeria [44]. The possible reason might be that rural women were less educated and less exposed to media messages [45]. Inaccessibility of services due to a lack of transportation and long distances to travel to the health institutions. Furthermore, rural women have less decision-making autonomy, less knowledge of pregnancy complications, and less access to information than urban women [23,46].

**Table 3. Multi-variable regression for factors associated with home delivery in Gidan District, Northeastern, Ethiopia, 2021(N = 760).**

| Variables | Home delivery | | COR(95%CI | AOR (95%CI) |
|---|---|---|---|---|
| | Yes | No | | |
| **Age (years)** | | | | |
| <20 | 25 | 49 | 1.16 (0.65–2.06) | 1.43(0.67–3.04) |
| 20–25 | 56 | 141 | 0.90 (0.58–1.14) | 0.89 (0.51–1.57) |
| 26–30 | 82 | 224 | 0.83 (0.55–1.24) | 0.86(0.53–1.38) |
| >30 | 56 | 127 | 1 | 1 |
| **Educational status respondents** | | | | |
| No formal Education | 111 | 235 | 2.05 (1.21–3.46) | 1.47 (0.78–2.79) |
| Primary | 87 | 215 | 1.75 (1.03–3.00) | 1.51 (0.81–2.29) |
| Secondary and above | 21 | 91 | 1 | 1 |
| **Education of husband** | | | | |
| No formal education | 118 | 250 | 1.98(1.38–2.84) | 2.60(1.72,3.91) |
| Formal education | 58 | 245 | 1 | 1 |
| **Residence** | | | | |
| Urban | 43 | 144 | 1 | 1 |
| Rural | 176 | 397 | 1.48 (1.01–2.18) | 2.02 (1.23–3.34)* |
| **Current marital status** | | | | |
| Married | 168 | 503 | 1 | 1 |
| Unmarried | 51 | 38 | 4.02 (2.55–6.33) | 11.16 (4.18–29.79)* |
| **Parity** | | | | |
| Primipara | 38 | 110 | 1 | 1 |
| Multipara | 181 | 431 | 1.22 (0.81–1.83) | 1.38 (0.79–2.38) |
| **Membership for WDA** | | | | |
| Yes | 36 | 120 | 1 | 1 |
| No | 183 | 421 | 1.45 (0.96–2.19) | 1.64 (1.01–2.65)* |
| **Fear of COVID-19** | | | | |
| No | 33 | 195 | 1 | 1 |
| Yes | 186 | 346 | 3.18 (2.11–479) | 3.86 (2.31–6.44)* |
| **Lack of transport** | | | | |
| No | 175 | 482 | 1 | 1 |
| Yes | 44 | 59 | 2.05 (1.34–3.15) | 1.54 (0.90–2.64)* |

**Notes**; *significantly associated; 1, reference.

**Abbreviations**; COR, crude odds ratio; AOR, adjusted odds ratio; COVID-19, coronavirus disease.

According to the findings of this study, unmarried women were 11.16 times more likely to deliver at home than married women. This finding is supported by other studies conducted in Jimma [47] and Southern Ethiopia [48]. The reason behind this might be that unmarried women do not receive support from husbands, and the health delivery system of Ethiopia favors married women over unmarried women [30].

Pregnant women whose husbands have no formal education have higher odds of delivering at home compared to those with formal education. This is consistent with studies reported in our country and abroad [19,20,22,49,50]. The possible explanation might be that non-educated husbands are more conservative towards the cultural practices of home delivery. Besides, non-educated husbands are unaware of the difficulties and complications associated with

pregnancy and childbirth. As a result, they are less likely to support their wives' use of maternal health care services [51,52]. It suggests that education is a major key strategy to increase maternal and child health care service utilization.

In this study, the odds of home delivery among women who were not involved in WDA were 1.65 times more likely to deliver at home as compared to women who were involved in WDA. This finding is incongruent with a study conducted in Southern Ethiopia [53]. The possible reason might be that women who are not part of WDA are less likely to discuss their health issues, including place of birth, among each other as well as with other health care providers. Moreover, scholars found that for women who live in an area with advanced WDA networks, the utilization of maternal health care services, including intuitional delivery, has greatly improved [28,29].

The findings of this study revealed that fear of COVID-19 infection was another factor that affected the place of delivery. Women who feared COVID-19 infection had 3.84 times higher odds of delivering at home than women who did not fear COVID-19 infection. This finding was consistent with a study done in India [32]. The reason behind this might be low awareness about the preventive measures for COVID-19. In addition, a study conducted in Italy among pregnant women found that there was a fear of visiting health institutions for delivery because of the COVID-19 infection [54].

## Strengths and limitations of the study

The main strength of the current study is that being a community-based study, it might reflect the actual experiences of the women during the study period. Even though all possible strategies were used, such as providing training for data collectors, employing pretests, and using standardized tools, the study might not be free of recall bias due to the data collected from women about their experiences one year ago. However, the cross-sectional nature of the study means that it cannot establish a cause-and-effect relationship. Another limitation of the current study is that, because it used only quantitative methods, some of the factors that affect the outcome variable may not have been addressed.

## Conclusion

Even though the government tried to lower the rate of home delivery by accessing health institutions in remote areas, implementing WDA, and introducing maternal waiting home utilization, nearly one in every three pregnant women had given birth at home among ANC-booked women in their last pregnancy. Factors such as being a rural resident, husband's education, unmarried women, not being involved in WDA, and fear of acquiring COVID-19 infection were found to be significant with home delivery. Improving the husband's educational status, providing information related to health institution delivery benefits during antenatal care, and strengthening the implementation of the women's development army, particularly among rural and unmarried women, would decrease home childbirth practices. Future researchers interested in the area should address why ANC-booked pregnant women preferred to deliver at home through a qualitative approach, which might have a tremendous effect on institutional service delivery utilization.

## Supporting information

**S1 File. Amharic version questionnaires.**
(DOCX)

**S2 File. English version questionnaire.**
(DOCX)

**S3 File. Data used for analysis including data on home delivery and associated factors.**
(XLS)

## Acknowledgments

We thank the University of Gondar for approving the ethical clearance. We are also pleased to extend our appreciation to the Gidan District administrative offices, study participants, data collectors, and supervisors.

## Author Contributions

**Conceptualization:** Desale Bihonegn Asmamaw.

**Data curation:** Desale Bihonegn Asmamaw, Eskedar Getie Mekonnen.

**Formal analysis:** Yohannes Ayanaw Habitu, Eskedar Getie Mekonnen, Wubshet Debebe Negash.

**Funding acquisition:** Desale Bihonegn Asmamaw.

**Investigation:** Yohannes Ayanaw Habitu.

**Methodology:** Desale Bihonegn Asmamaw.

**Software:** Desale Bihonegn Asmamaw, Yohannes Ayanaw Habitu, Eskedar Getie Mekonnen.

**Supervision:** Eskedar Getie Mekonnen.

**Visualization:** Wubshet Debebe Negash.

**Writing – original draft:** Desale Bihonegn Asmamaw.

**Writing – review & editing:** Yohannes Ayanaw Habitu, Eskedar Getie Mekonnen, Wubshet Debebe Negash.

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
