## [Decision Letter · Decision Letter 0]

21 Jun 2022

PONE-D-22-10889Home delivery and associated factors among antenatal care booked women in their last pregnancy in Northeast EthiopiaPLOS ONE

Dear Dr.Desale Bihonegn Asmamaw ,

Thank you for submitting your manuscript to PLOS ONE. After careful consideration, we feel that it has merit but does not fully meet PLOS ONE’s publication criteria as it currently stands. Therefore, we invite you to submit a revised version of the manuscript that addresses the points raised during the review process.

We look forward to receiving your revised manuscript.

Kind regards,

Pracheth Raghuveer, MD, DNB

Academic Editor

PLOS ONE

Journal Requirements:

"NA"

"No funding available"

Additional Editor Comments:

Dear author,

Please revise the manuscript as per the comments made by the reviewers.

Reviewers' comments:

Reviewer's Responses to Questions

**Comments to the Author**

1. Is the manuscript technically sound, and do the data support the conclusions?

Reviewer #1: Yes

Reviewer #2: No

2. Has the statistical analysis been performed appropriately and rigorously? 

Reviewer #1: Yes

Reviewer #2: No

3. Have the authors made all data underlying the findings in their manuscript fully available?

Reviewer #1: Yes

Reviewer #2: No

4. Is the manuscript presented in an intelligible fashion and written in standard English?

Reviewer #1: No

Reviewer #2: Yes

5. Review Comments to the Author

Reviewer #1: Background section-Page 4, Line 51

Of this two-thirds of the deaths from SSA

I think you mean; Of this two-thirds of the deaths were from SSA

Page 5, line 76,77; Another studies in Ethiopia revealed that many women 77 unpredictably did not deliver in health facilities despite antenatal care

You should write as Another study from Ethiopia revealed that many women 77 unpredictably did not deliver in health facilities despite antenatal care

Study variables-Page 7, Line 121, 122

When a women gave birth at her home or others’ home 122 (neighbor, relatives, or family) or when a birth takes place outsides of health institutions.

You should write, When a woman gave birth at her home or others’ home 122 (neighbor, relatives, or family) or when a birth took place outsides of health institutions

Result section-Page 10, Line 166, 167

306 (40.3%) of participants fall within the age category of 167 26-30 years.

You should write, 306 (40.3%) of participants fell within the age category of 167 26-30 years.

Page 10, Line 172, 173

About 532(70%) of the mothers were feared 173 COVID-19 infection

You should write, About 532(70%) of the mothers feared 173 COVID-19 infection

Discussion-Pg 11, 12, Line-194, 195

a significant number of pregnant women still give birth at home, contrary to plane of health institution delivery.

You should write-a significant number of pregnant women still give birth at home, contrary to plan of health institution delivery.

Pg 12, Line-199, 200

This difference might be due to the effect of COVID-19 pandemic at which women had high-perceived severity.

Rephrase this line

Pg 12, Line-206-209

This lower prevalence might be that time and study population difference as the current study included only those individuals who had ANC follow up which is one of the most known determinants of home delivery and the government’s ongoing effort to improve the health care system

Rephrase this statement

Pg 12, Line-212, 213

This implied that strengthening the above activities will be resulted a better outcome.

You should write-This implied that strengthening the above activities will result in a better outcome.

Pg 13, Line-218-220

Moreover, women form rural areas were limited decision-making autonomy, lack of knowledge of pregnancy complication and limited access to information than urban women

You should write-Moreover, women from rural areas with limited decision-making autonomy, lack of knowledge of pregnancy complications, and limited access to information from urban women

Pg 13, Line-229, 230

Besides, non-educated husbands do not aware the difficulty and complication happen during pregnancy and childbirth.

You should write-Besides, non-educated husbands are unaware of the difficulty and complications associated with pregnancy and childbirth.

Pg 13, Line-234, 235

This finding 235 is in congruent with a study conducted in Southern Ethiopia [47],

Write a full stop at the end.

Pg 13, Line-235- 237

The possible reason might be women who did not involve in WDA are less likely to discuss with health care providers and with each other about their health including where to give birth

The possible reason might be that women who are not part of WDA are less likely to discuss their health issues, including place of birth among each other as well as other health care providers.

Pg 14, Line-240-246

The finding of this study revealed that fear of COVID-19 infection were another factor that affect place of delivery. Women who were feared of COVID-19 infection were 3.84 times more odds to deliver at home than women who were not feared COVID-19 infection, this finding was consistent with a study done in India. The reason behind might be low awareness about the preventive measures of COVID-19, in addition, a study conducted in Italy among pregnant women found that there were fear of visiting health facilities for delivery because of fearing COVID-19 infection

You should write-The finding of this study revealed that fear of COVID-19 infection was another factor that affected the place of delivery. Women who feared COVID-19 infection had 3.84 times more odds to deliver at home than women who did not fear COVID-19 infection. This finding was consistent with a study done in India. The reason behind this might be low awareness about the preventive measures for COVID-19. In addition, a study conducted in Italy among pregnant women found that there was fear of visiting health facilities for delivery because of COVID-19 infection.

Strength and limitations-Pg 14, Line 250, 251

In addition, May introduce a 251 recall bias due to data was collected from women about their experience since 1 year back.

You should write-In addition, there might be a recall bias due to data collected from women about their experiences 1 year back.

Conclusion-Pg 14, Line-258, 259

improving the husband’s educational status, providing information related to health institution delivery benefits

You should write-Improving the husband’s educational status, providing information related to health institution delivery benefits

Page-14, 15, Line-261-264

Future researchers interested in the area better to address why pregnant women preferred to deliver in the home though qualitative approach, which might have a tremendous effect on institutional services delivery utilization.

You should write-Future researchers interested in the area better address why pregnant women preferred to deliver in the home through a qualitative approach, which might have a tremendous effect on institutional service delivery utilization.

I think you should look into table 3-home delivery variable. Were there no home deliveries as there are no numbers mentioned under the YES home-delivery variable?

Reviewer #2: Paragraph 2, page five mentions: While home delivery is usually the cheapest option in resource poor settings like Ethiopia, it is also the most risky, due to the possibility of infection or lack of equipment in case of complications arise.

Comment:

Please substantiate why this is the cheapest option? What if any measures are taken by the government/NGOs to help overcome this to enable institutional deliveries? Are there any rates of how many women in the study area do access ANC?

2. The fear of COVID-19 was assessed. However, it is not clear if this was in relation to acquiring the disease per se during pregnancy or was pertinent in the context of accessing healthcare services during delivery and contracting the disease. Please clarify for reader's comprehension.

3. Paragraph 1, page 12, line 201: 'In addition, possibly due to 201 internal conflicts in the area.' This statement does not seem complete. Also, this seems to be an important determinant which may preclude women from accessing delivery services.

4. In the conclusion, particular focus needs to also be given to the fact that one of the highest odds of home deliveries were among unmarried women. There is mention elsewhere that married women were given preference at the health centers. An educational program targeting healthcare workers to work with empathy and to discharge their duties without bias and/or discrimination may encourage more women to access these services and consequently, reduce maternal and infant morbidity and mortality. If such provisions are already in place through the WDA, they can be highlighted as well.

5. Please clarify if in Table 1 the 'single' mothers under marital status implies 'unmarried women'? Also, the next variable needs to be corrected to 'widowed' as it is at present, 'windowed.' A thorough grammar and spell check is suggested.

6. Married women in Table 1 is 684 but in table 3 is 671. Why the discrepancy? Also, were divorced and separated women clubbed with unmarried women? Please clarify. Please check all variables.

7. There seem to be differences in the frequency between tables. Eg: Marital status variable in Table 1 does not match that in table 3.

6. PLOS authors have the option to publish the peer review history of their article (what does this mean?). If published, this will include your full peer review and any attached files.

Reviewer #1: **Yes: **Dr Saida Abrar

Reviewer #2: No

---

## [Author Response · Author response to Decision Letter 0]

2 Jul 2022

Authors’ response to the editor and reviewer’s comments

Title: Home delivery and associated factors among antenatal care booked women in their last pregnancy in Northeast Ethiopia.

Authors 

Desale Bihonegn Asmamaw (desalebihonegn1988@gmail.com)

Yohannes Ayanaw Habitu ((yohaneshabitu@gmail.com)

Eskedar Getie Mekonnen (eskedargetie18@gmail.com)

Wubshet Debebe Negash (wubshetdn@gmail.com)

Date: 28/06/2022

Authors’ Point-by-point response to editor and reviewer comments 

We are very grateful to both the editor and reviewers for your comments and concerns. All the concerns raised so far will have an undeniable impact on improving the quality and readability of our scholarly work. Appreciating your effort and valuable comments, we have provided possible reflections on the raised concerns and questions. Kindly find our reflections here.

S. no. Comments 

 Reviewer 1 comments 

1 Background section-Page 4, Line 51

Of this two-thirds of the deaths from SSA

I think you mean; Of this two-thirds of the deaths were from SSA 

Authors response 

Thank you for your observations. This has been corrected. Kindly see page 4, line 51-52.

2 Page 5, line 76,77; Another study in Ethiopia revealed that many women 77 unpredictably did not deliver in health facilities despite antenatal care

You should write as Another study from Ethiopia revealed that many women 77 unpredictably did not deliver in health facilities despite antenatal care 

Authors response 

Comment accepted. This has been modified. kindly see page 5, line 76-77. 

3 Study variables-Page 7, Line 121, 122

When a women gave birth at her home or others’ home 122 (neighbor, relatives, or family) or when a birth takes place outsides of health institutions.

You should write, When a woman gave birth at her home or others’ home 122 (neighbor, relatives, or family) or when a birth took place outsides of health institutions 

Authors response 

Thank you for your observation. This has been corrected. Kindly see page 7, line 121-124.

4 Result section-Page 10, Line 166, 167

306 (40.3%) of participants fall within the age category of 167 26-30 years.

You should write, 306 (40.3%) of participants fell within the age category of 167 26-30 years. Thank you for your comments and your comment is accepted. Kindly see page 10, line 175-177.

5 Page 10, Line 172, 173

About 532(70%) of the mothers were feared 173 COVID-19 infection

You should write, About 532(70%) of the mothers feared 173 COVID-19 infection Thank you for your observations. This has been addressed. Kindly see page 10, line 182-83.

6 Discussion-Pg 11, 12, Line-194, 195

a significant number of pregnant women still give birth at home, contrary to plane of health institution delivery.

You should write-a significant number of pregnant women still give birth at home, contrary to plan of health institution delivery. Comment accepted. The mistake has been corrected. Kindly see page 11, line 203-204.

7 Pg 12, Line-199, 200

This difference might be due to the effect of COVID-19 pandemic at which women had high-perceived severity.

Rephrase this line Thank you for your comments and your comment is accepted. This has been addressed accordingly. Kindly see page 12, line 210-214.

8 g 12, Line-206-209

This lower prevalence might be that time and study population difference as the current study included only those individuals who had ANC follow up which is one of the most known determinants of home delivery and the government’s ongoing effort to improve the health care system

Rephrase this statement Thank you for your suggestions. This has been rephrased. Kindly see page 12. Line 220-223

9 Pg 12, Line-212, 213

This implied that strengthening the above activities will be resulted a better outcome.

You should write-This implied that strengthening the above activities will result in a better outcome. Thank you for your observations. This has been addressed. Kindly see page 12, line 226-227.

10 Pg 13, Line-218-220

Moreover, women form rural areas were limited decision-making autonomy, lack of knowledge of pregnancy complication and limited access to information than urban women

You should write-Moreover, women from rural areas with limited decision-making autonomy, lack of knowledge of pregnancy complications, and limited access to information from urban women Thank you for your suggestions. This has been modified accordingly. Page 13, line 232-234.

11 Pg 13, Line-229, 230

Besides, non-educated husbands do not aware the difficulty and complication happen during pregnancy and childbirth.

You should write-Besides, non-educated husbands are unaware of the difficulty and complications associated with pregnancy and childbirth. Thank you for your comments. This has been addressed. Kindly see page 13, line 243-245.

12 Pg 13, Line-234, 235

This finding 235 is in congruent with a study conducted in Southern Ethiopia [47],

Write a full stop at the end. Thank you for your observations. The grammatical error has been addressed, kindly see page 13, line 249-250.

13 Pg 13, Line-235- 237

The possible reason might be women who did not involve in WDA are less likely to discuss with health care providers and with each other about their health including where to give birth

The possible reason might be that women who are not part of WDA are less likely to discuss their health issues, including place of birth among each other as well as other health care providers. 

14 Pg 14, Line-240-246

The finding of this study revealed that fear of COVID-19 infection were another factor that affect place of delivery. Women who were feared of COVID-19 infection were 3.84 times more odds to deliver at home than women who were not feared COVID-19 infection, this finding was consistent with a study done in India. The reason behind might be low awareness about the preventive measures of COVID-19, in addition, a study conducted in Italy among pregnant women found that there were fear of visiting health facilities for delivery because of fearing COVID-19 infection

You should write-The finding of this study revealed that fear of COVID-19 infection was another factor that affected the place of delivery. Women who feared COVID-19 infection had 3.84 times more odds to deliver at home than women who did not fear COVID-19 infection. This finding was consistent with a study done in India. The reason behind this might be low awareness about the preventive measures for COVID-19. In addition, a study conducted in Italy among pregnant women found that there was fear of visiting health facilities for delivery because of COVID-19 infection. Thank you for your suggestions. This has been addressed. Kindly see page 14, line 255-260.

15 Strength and limitations-Pg 14, Line 250, 251

In addition, May introduce a 251 recall bias due to data was collected from women about their experience since 1 year back.

You should write-In addition; there might be a recall bias due to data collected from women about their experiences 1 year back. Thank you for your suggestions. We have corrected it. Kindly see page 14, line 263-265.

16 Conclusion-Pg 14, Line-258, 259

improving the husband’s educational status, providing information related to health institution delivery benefits

You should write-Improving the husband’s educational status, providing information related to health institution delivery benefits Thank you for your observations. The grammar error has been corrected. Kindly see page 14, line 272-273.

17 Page-14, 15, Line-261-264

Future researchers interested in the area better to address why pregnant women preferred to deliver in the home though qualitative approach, which might have a tremendous effect on institutional services delivery utilization.

You should write-Future researchers interested in the area better address why pregnant women preferred to deliver in the home through a qualitative approach, which might have a tremendous effect on institutional service delivery utilization. Thank you for your comments. This has been modified accordingly. Kindly see page 15, line 280-283.

18 I think you should look into table 3-home delivery variable. Were there no home deliveries as there are no numbers mentioned under the YES home-delivery variable? Thank you for you observations. This has been addressed. Kindly see table 3.

 Reviewer 2 comments Authors response 

1 Please substantiate why this is the cheapest option? What if any measures are taken by the government/NGOs to help overcome this to enable institutional deliveries? Are there any rates of how many women in the study area do access ANC? Thank you for your comments; we modified it according to our research questions. Kindly see page 5, line 72-74.

2 The fear of COVID-19 was assessed. However, it is not clear if this was in relation to acquiring the disease per se during pregnancy or was pertinent in the context of accessing healthcare services during delivery and contracting the disease. Please clarify for reader's comprehension. Comment accepted. This has been addressed. Kindly see page 7, line 128-130. 

3 Paragraph 1, page 12, line 201: 'In addition, possibly due to 201 internal conflicts in the area.' This statement does not seem complete. Also, this seems to be an important determinant which may preclude women from accessing delivery services. Comment accepted and necessary modifications have been considered. Kindly see page 12, line 202-208.

4 In the conclusion, particular focus needs to also be given to the fact that one of the highest odds of home deliveries were among unmarried women. There is mention elsewhere that married women were given preference at the health centers. An educational program targeting healthcare workers to work with empathy and to discharge their duties without bias and/or discrimination may encourage more women to access these services and consequently, reduce maternal and infant morbidity and mortality. If such provisions are already in place through the WDA, they can be highlighted as well. Thank you for your suggestions, this has been addressed and rewritten. See the updated version of the manuscript.

5 Please clarify if in Table 1 the 'single' mothers under marital status implies 'unmarried women'? Also, the next variable needs to be corrected to 'widowed' as it is at present, 'windowed.' A thorough grammar and spell check is suggested. Thank you for your observations. The spelling mistake has been corrected. Single mothers were categorized as unmarried women [1, 2].

6 Married women in Table 1 is 684 but in table 3 is 671. Why the discrepancy? Also, were divorced and separated women clubbed with unmarried women? Please clarify. Please check all variables. Thank you for your observations. The difference in the marital status frequency between tables 1 and 3 is a typing error. The mistake has been corrected. We categorized the marital status based on the current marital status (marital status at the time of the data collection). So, divorced and separated women clubbed with unmarried women. Kindly see the updated version of the manuscript.

7 There seem to be differences in the frequency between tables. Eg: Marital status variable in Table 1 does not match that in table 3. Comment accepted. These have been typing errors and the mistake has been corrected in the updated version of the manuscript.

Reference 

1. Fikre AA, Demissie M: Prevalence of institutional delivery and associated factors in Dodota Woreda (district), Oromia regional state, Ethiopia. Reproductive health 2012, 9(1):1-6.

2. Teshale AB, Tesema GA: Prevalence and associated factors of delayed first antenatal care booking among reproductive age women in Ethiopia; a multilevel analysis of EDHS 2016 data. PloS one 2020, 15(7):e0235538.

 Thank you very much!

---

## [Decision Letter · Decision Letter 1]

3 Aug 2022

PONE-D-22-10889R1Home delivery and associated factors among antenatal care booked women in their last pregnancy in Northeast EthiopiaPLOS ONE

Dear Dr. Desale Bihonegn Asmamaw,

Thank you for submitting your manuscript to PLOS ONE. After careful consideration, we feel that it has merit but does not fully meet PLOS ONE’s publication criteria as it currently stands. Therefore, we invite you to submit a revised version of the manuscript that addresses the points raised during the review process.

The reviewers think that the manuscript needs further revision. . 

We look forward to receiving your revised manuscript.

Kind regards,

Pracheth Raghuveer, MD, DNB

Academic Editor

PLOS ONE

Reviewers' comments:

Reviewer's Responses to Questions

**Comments to the Author**

1. If the authors have adequately addressed your comments raised in a previous round of review and you feel that this manuscript is now acceptable for publication, you may indicate that here to bypass the “Comments to the Author” section, enter your conflict of interest statement in the “Confidential to Editor” section, and submit your "Accept" recommendation.

Reviewer #1: (No Response)

2. Is the manuscript technically sound, and do the data support the conclusions?

Reviewer #1: Yes

3. Has the statistical analysis been performed appropriately and rigorously? 

Reviewer #1: Yes

4. Have the authors made all data underlying the findings in their manuscript fully available?

Reviewer #1: Yes

5. Is the manuscript presented in an intelligible fashion and written in standard English?

Reviewer #1: No

6. Review Comments to the Author

Reviewer #1: dear author

You have failed to address all my previous queries which were previously highlighted. Kindly address them pointwise and make sure you enter them in details.

Background section-Page 4, Line 51

Of this two-thirds of the deaths from SSA

I think you mean; Of this two-thirds of the deaths were from SSA

Page 5, line 76,77; Another studies in Ethiopia revealed that many women 77 unpredictably did not deliver in health facilities despite antenatal care

You should write as Another study from Ethiopia revealed that many women 77 unpredictably did not deliver in health facilities despite antenatal care

Study variables-Page 7, Line 121, 122

When a women gave birth at her home or others’ home 122 (neighbor, relatives, or family) or when a birth takes place outsides of health institutions.

You should write, When a woman gave birth at her home or others’ home 122 (neighbor, relatives, or family) or when a birth took place outsides of health institutions

Result section-Page 10, Line 166, 167

306 (40.3%) of participants fall within the age category of 167 26-30 years.

You should write, 306 (40.3%) of participants fell within the age category of 167 26-30 years.

Page 10, Line 172, 173

About 532(70%) of the mothers were feared 173 COVID-19 infection

You should write, About 532(70%) of the mothers feared 173 COVID-19 infection

Discussion-Pg 11, 12, Line-194, 195

a significant number of pregnant women still give birth at home, contrary to plane of health institution delivery.

You should write-a significant number of pregnant women still give birth at home, contrary to plan of health institution delivery.

Pg 12, Line-199, 200

This difference might be due to the effect of COVID-19 pandemic at which women had high-perceived severity.

Rephrase this line

Pg 12, Line-206-209

This lower prevalence might be that time and study population difference as the current study included only those individuals who had ANC follow up which is one of the most known determinants of home delivery and the government’s ongoing effort to improve the health care system

Rephrase this statement

Pg 12, Line-212, 213

This implied that strengthening the above activities will be resulted a better outcome.

You should write-This implied that strengthening the above activities will result in a better outcome.

Pg 13, Line-218-220

Moreover, women form rural areas were limited decision-making autonomy, lack of knowledge of pregnancy complication and limited access to information than urban women

You should write-Moreover, women from rural areas with limited decision-making autonomy, lack of knowledge of pregnancy complications, and limited access to information from urban women

Pg 13, Line-229, 230

Besides, non-educated husbands do not aware the difficulty and complication happen during pregnancy and childbirth.

You should write-Besides, non-educated husbands are unaware of the difficulty and complications associated with pregnancy and childbirth.

Pg 13, Line-234, 235

This finding 235 is in congruent with a study conducted in Southern Ethiopia [47],

Write a full stop at the end.

Pg 13, Line-235- 237

The possible reason might be women who did not involve in WDA are less likely to discuss with health care providers and with each other about their health including where to give birth

The possible reason might be that women who are not part of WDA are less likely to discuss their health issues, including place of birth among each other as well as other health care providers.

Pg 14, Line-240-246

The finding of this study revealed that fear of COVID-19 infection were another factor that affect place of delivery. Women who were feared of COVID-19 infection were 3.84 times more odds to deliver at home than women who were not feared COVID-19 infection, this finding was consistent with a study done in India. The reason behind might be low awareness about the preventive measures of COVID-19, in addition, a study conducted in Italy among pregnant women found that there were fear of visiting health facilities for delivery because of fearing COVID-19 infection

You should write-The finding of this study revealed that fear of COVID-19 infection was another factor that affected the place of delivery. Women who feared COVID-19 infection had 3.84 times more odds to deliver at home than women who did not fear COVID-19 infection. This finding was consistent with a study done in India. The reason behind this might be low awareness about the preventive measures for COVID-19. In addition, a study conducted in Italy among pregnant women found that there was fear of visiting health facilities for delivery because of COVID-19 infection.

Strength and limitations-Pg 14, Line 250, 251

In addition, May introduce a 251 recall bias due to data was collected from women about their experience since 1 year back.

You should write-In addition, there might be a recall bias due to data collected from women about their experiences 1 year back.

Conclusion-Pg 14, Line-258, 259

improving the husband’s educational status, providing information related to health institution delivery benefits

You should write-Improving the husband’s educational status, providing information related to health institution delivery benefits

Page-14, 15, Line-261-264

Future researchers interested in the area better to address why pregnant women preferred to deliver in the home though qualitative approach, which might have a tremendous effect on institutional services delivery utilization.

You should write-Future researchers interested in the area better address why pregnant women preferred to deliver in the home through a qualitative approach, which might have a tremendous effect on institutional service delivery utilization.

I think you should look into table 3-home delivery variable. Were there no home deliveries as there are no numbers mentioned under the YES home-delivery variable?

7. PLOS authors have the option to publish the peer review history of their article (what does this mean?). If published, this will include your full peer review and any attached files.

Reviewer #1: **Yes: **SAIDA ABRAR

---

## [Decision Letter · Decision Letter 2]

16 Nov 2022

PONE-D-22-10889R2Home delivery and associated factors among antenatal care booked women in their last pregnancy in Northeast EthiopiaPLOS ONE

Dear Dr. Asmamaw

Thank you for submitting your manuscript to PLOS ONE. After careful consideration, we feel that it has merit but does not fully meet PLOS ONE’s publication criteria as it currently stands. Therefore, we invite you to submit a revised version of the manuscript that addresses the points raised during the review process.

Kindly address the reviewer's comments

We look forward to receiving your revised manuscript.

Kind regards,

Pracheth Raghuveer, MD, DNB

Academic Editor

PLOS ONE

Journal Requirements:

Reviewers' comments:

Reviewer's Responses to Questions

**Comments to the Author**

1. If the authors have adequately addressed your comments raised in a previous round of review and you feel that this manuscript is now acceptable for publication, you may indicate that here to bypass the “Comments to the Author” section, enter your conflict of interest statement in the “Confidential to Editor” section, and submit your "Accept" recommendation.

Reviewer #1: All comments have been addressed

Reviewer #3: (No Response)

2. Is the manuscript technically sound, and do the data support the conclusions?

Reviewer #1: Yes

Reviewer #3: Yes

3. Has the statistical analysis been performed appropriately and rigorously? 

Reviewer #1: Yes

Reviewer #3: Yes

4. Have the authors made all data underlying the findings in their manuscript fully available?

Reviewer #1: Yes

Reviewer #3: Yes

5. Is the manuscript presented in an intelligible fashion and written in standard English?

Reviewer #1: Yes

Reviewer #3: Yes

6. Review Comments to the Author

Reviewer #1: All queries are effectively addressed and i recommend it for publication

The highlighted grammatical mistakes in the introduction, methodology, result sections are rectified.

All the journal requirements are satisfied and met

Reviewer #3: Dear Plos ONE team of editorials, thank you for the chance given to me to review a manuscript titled “Home delivery and associated factors among antenatal care booked women in their last 2 pregnancy in Northeast Ethiopia”. The following are my comments;

A. General Comments

• Refine the title. The title should not at least briefly explain the whole content of the manuscript. You can use absorbing and attractive findings to be your title.

•

• Tense flaws i.e., the background section and the result section.

• Incorrect use of words E.g., the “…was delivered” can be written as “…had given birth”

• Update the background with new initiatives. E.g. MOH new ANC guideline and other maternal initiatives of the ministry.

• You can frame the whole section of the manuscript “…. discontinuity in the continuum of maternal health care”

• Tense, grammar, content of the sections, presentation needs further emphasis.

• Paragraph should not be one sentence.

B. Specific Comments

• What is the motive behind booking of ANC? Why the book first and then fail to trust the institutional delivery? Would you explain it before and after the introduction of COVID 19?

• Differentiate on your literature between the ANC full care and Last booked ANC care throughout the document.

• Where did you include in the outcome if a mother give birth on her way to health facility?

• What is known about booking and then leaving the continuum of care? What is unknown? And what do you want to fill by this study?

• Why multi stage for a district? The sampling procedure is not clear to the reader. Is that multi stage? Is that stratified? Is that SRS?

• Why don’t you include the three delays?

• You had visited three times. What is the day time you have visited the home?

• The data quality assurance should be clearer at the three stage of the process and how it was ensured?

• When did they collect the data? Is that immediately at giving birth, or after one month of their giving birth…etc?

• Confidence interval should follow your percentage?

• Why the result section was narrower than the discussion section?

• Revisit for the contents of the sections in general

• Revisits the statistics again for frequency, univariate, binary and multiple logistic regression e.g. Why P value < 0.05? P value < 0.025?

• Finally, you can cross tabulate women who fear infection of COVID 19 and home delivery. Why? You can even reframe the title by this.

Regards,

7. PLOS authors have the option to publish the peer review history of their article (what does this mean?). If published, this will include your full peer review and any attached files.

Reviewer #1: **Yes: **Saida Abrar

Reviewer #3: No

---

## [Author Response · Author response to Decision Letter 2]

20 Nov 2022

Authors’ response to the editor and reviewer’s comments

Title: Impact of fear of COVID-19 infection on home delivery among antenatal care booked women in Northeast Ethiopia

Authors 

Desale Bihonegn Asmamaw (desalebihonegn1988@gmail.com)

Yohannes Ayanaw Habitu ((yohaneshabitu@gmail.com)

Eskedar Getie Mekonnen (eskedargetie18@gmail.com)

Wubshet Debebe Negash (wubshetdn@gmail.com)

Date: 20/112022

Authors’ Point-by-point response to editor and reviewer comments 

Dear reviewer, we would like to extend our deepest appreciation for devoting your time to reviewing our manuscript entitled “impact of fear of COVID-19 infection on home delivery among antenatal care booked women in Northeast Ethiopia ". There has been a major revision of this manuscript (Abstract, introduction, methods, results, discussion, and conclusions). The whole structure of the manuscript has been revised. We hope now the manuscript is clear and more acceptable than its previous version. We have tried to present the paper in the proper manner according to your comment on what to suppose to do so. For this, here we have given our responses to each of the concerns you raised. Again, we would like to remind our strongest gratitude for your effort in the improvement of this manuscript, and all the points were addressed in the point-by-point response.

General comments of the reviewer

Reviewer comments: Refine the title. The title should not at least briefly explain the whole content of the manuscript. You can use absorbing and attractive findings to be your title.

Authors’ response: Thank you for your suggestion, this has been modified according to the findings, kindly see the updated version of the title.

Reviewer comments: Tense flaws i.e., the background section and the result section.

Authors’ response: Thank you for your comments, this has been edited, kindly see the updated version of the manuscript.

Reviewer comments: Incorrect use of words E.g., the “…was delivered” can be written as “…had given birth”

Authors response: Thank you for your observations, this has been addressed. Kindly see the updated version of the manuscript.

Reviewer comments: Update the background with new initiatives. E.g. MOH new ANC guideline and other maternal initiatives of the ministry.

Authors’ response: Thank you for your comments, this has been addressed. Kindly see the updated version of the introduction. 

Reviewer comments: You can frame the whole section of the manuscript “…. discontinuity in the continuum of maternal health care

Authors response: Dear reviewer, thank you for the comments and suggestions, this has been modified accordingly. Kindly see the updated version of the manuscript.

Reviewer comments: Tense, grammar, the content of the sections, and presentation needs further emphasis.

Authors response: Thank you for your observation and suggestions. The entire article has been read and corrected by a native English speaker. Kindly see the updated version of the manuscript. 

Reviewer comments: Paragraph should not be one sentence.

Authors response: Thank you for your observations and suggestions. This has been addressed; kindly see the updated version of the manuscript.

Specific comments 

Reviewer comments: What is the motive behind booking of ANC? Why the book first and then fail to trust the institutional delivery? Would you explain it before and after the introduction of COVID 19?

Authors response: Thank you for the comments and observations. In fact, pregnant women who had ANC visits are expected to give birth at health institutions, there are studies that found that pregnant women who had ANC visits gave birth at home (drop out from the continuum of maternal health care) [1, 2]. Home birth after ANC is a major public health issue in Ethiopia, despite the fact that few studies on the determinants of the practice have been conducted [1]. We tried to explain it before and after the introduction of COVID-19. Kindly the updated version of the introduction and limitation section.

Reviewer comments: Differentiate your literature between the ANC full care and Last booked ANC care throughout the document.

Authors response: Thank you for your suggestion, this has been differentiated. Kindly see the updated version of the document.

Reviewer comments: Where did you include in the outcome if a mother give birth on her way to health facility?

Authors response: Home delivery was defined as any birth that had taken place in the woman’s or others’ home, while deliveries that occurred in governmental health posts, health centers, hospitals, private clinics and hospitals, and NG health facilities were grouped as facility-based deliveries. If a mother gives birth on her way to health facilities, she was considered a home delivery [2].

Reviewer comments: What is known about booking and then leaving the continuum of care? What is unknown? And what do you want to fill by this study?

Authors response: Ethiopia strives to end maternal and neonatal mortalities through increased production of skilled professionals on maternal and child health, collaboration with different governmental and non-governmental organizations, increased budget allocation, and give special emphasis. In Ethiopia according to the 2016 EDHS report, 62% of women were booked for ANC and only 26% of pregnant women had given birth at health institutions. There are variety of literature reporting factors that affect the continuum of maternal health care [2, 4-10], but there is no adequate literature explaining why ANC-booked women prefer home deliveries and the impact of fear of COVID-19 infection on home delivery. This leaves maternal morbidity and mortality as a public health problem. Therefore, the aim of this study was to assess the impact of fear of COVID-19 on home delivery among women who had booked ANC. Kindly see the updated version of the introduction.

Reviewer comments: Why multi stage for a district? The sampling procedure is not clear to the reader. Is that multi stage? Is that stratified? Is that SRS?

Authors response: Multistage sampling was employed to select study subjects. First, all kebeles in the District were stratified in to urban and rural. From 23 kebeles stratified into two (21 rural kebeles and two urban kebele), six rural and one urban kebeles were selected by simple random sampling technique using the lottery method. Mothers in the sampled kebeles selected by using simple random sampling technique (open Epi Random Program version 3) [11-13]. Kindly see page 8, line 119-130, and figure 1.

Reviewer comments: You had visited three times. What is the day time you have visited the home?

Authors’ response: Dear reviewer, thank you for your comments. The data was collected from home to home; the data collectors visited the homes on working days (Monday to Friday) because, in the Ethiopian context, almost all rural women and most urban women go to church (mostly orthodox) and market on Saturday and Sunday. The data collectors returned to the home if they did not find the participants at the time. When the interviewers failed to find the eligible respondent after three visits, the next household was included.

Reviewer comments: The data quality assurance should be clearer at the three stage of the process and how it was ensured?

Authors response: Thank you for the comments and suggestions, this has been addressed in detail, kindly see page 9, line 143-152.

Reviewer comments: When did they collect the data? Is that immediately at giving birth, or after one month of their giving birth…etc?

Authors response: Thank you for your comments. Regardless of the outcome, limiting the participants to only a 1-year period and after one month of giving birth was made to minimize potential recall bias and make the mother stabilized and comfortable. Kindly see page 8, line 127-130.

Reviewer comments: Confidence interval should follow your percentage?

Authors response: thank you for your comments, this has been dressed kindly see the updated version of the manuscript.

Reviewer comments: Revisit for the contents of the sections in general

Reviewer comments: Thank you for your suggestions. We were revisited the contents of the section. Kindly see the updated version of the manuscript.

Reviewer comments: Revisits the statistics again for frequency, univariate, binary and multiple logistic regression e.g. Why P value < 0.05? P value < 0.025?

Authors response: variables at less than or equals to 0.25 p-values in the bivariable logistic regression model were fitted into the multivariable logistic regression model to control the effect of confounding variables. Variable having P-value less than or equals to 0.05 in the multivariable logistic regression analysis was considered as associated factors for factors for home delivery. The reason for using P value < 0.05? P value < 0.25? Based on reviewing literatures and there are a lot of research that are used such kinds of cut of point [2, 11, 13]. Kindly consider these kinds of issues.

Reviewer comments: Finally, you can cross tabulate women who fear infection of COVID 19 and home delivery. Why? You can even reframe the title by this

Authors response: Thank you for your comments and suggestions, this has been addressed according. Kindly see the updated version of the manuscript.

1. Muluneh AG, Animut Y, Ayele TA: Spatial clustering and determinants of home birth after at least one antenatal care visit in Ethiopia: Ethiopian demographic and health survey 2016 perspective. BMC pregnancy and childbirth 2020, 20(1):1-13.

2. Kasaye HK, Endale ZM, Gudayu TW, Desta MS: Home delivery among antenatal care booked women in their last pregnancy and associated factors: community-based cross sectional study in Debremarkos town, North West Ethiopia, January 2016. BMC pregnancy and childbirth 2017, 17(1):1-12.

3. Berhan Y, Berhan A: Antenatal care as a means of increasing birth in the health facility and reducing maternal mortality: a systematic review. Ethiopian journal of health sciences 2014, 24:93-104.

4. Wodaynew T, Fekecha B, Abdisa B: Magnitude of home delivery and associated factors among antenatal care booked mothers in Delanta District, South Wollo Zone, North East Ethiopia: a cross-sectional study, March 2018. Int J Womens Health Wellness 2018, 4(2):1-11.

5. Tessema ZT, Tiruneh SA: Spatio-temporal distribution and associated factors of home delivery in Ethiopia. Further multilevel and spatial analysis of Ethiopian demographic and health surveys 2005–2016. BMC pregnancy and childbirth 2020, 20(1):1-16.

6. Kifle MM, Kesete HF, Gaim HT, Angosom GS, Araya MB: Health facility or home delivery? Factors influencing the choice of delivery place among mothers living in rural communities of Eritrea. Journal of Health, Population and Nutrition 2018, 37(1):1-15.

7. Delibo D, Damena M, Gobena T, Balcha B: Status of home delivery and its associated factors among women who gave birth within the last 12 months in east Badawacho District, Hadiya zone, Southern Ethiopia. BioMed Research International 2020, 2020.

8. Chernet AG, Dumga KT, Cherie KT: Home delivery practices and associated factors in Ethiopia. Journal of reproduction & infertility 2019, 20(2):102.

9. Amano A, Gebeyehu A, Birhanu Z: Institutional delivery service utilization in Munisa Woreda, South East Ethiopia: a community based cross-sectional study. BMC pregnancy and childbirth 2012, 12(1):1-6.

10. Ayele G, Tilahune M, Merdikyos B, Animaw W, Taye W: Prevalence and associated factors of home delivery in Arbaminch Zuria district, southern Ethiopia: community based cross sectional study. Science 2015, 3(1):6-9.

11. Tsegaye AT, Mengistu M, Shimeka A: Prevalence of unintended pregnancy and associated factors among married women in west Belessa Woreda, Northwest Ethiopia, 2016. Reproductive health 2018, 15(1):1-8.

12. Markos D, Bogale D: Birth preparedness and complication readiness among women of child bearing age group in Goba woreda, Oromia region, Ethiopia. BMC pregnancy and childbirth 2014, 14(1):1-9.

13. Yisak H, Gobena T, Mesfin F: Prevalence and risk factors for under nutrition among children under five at Haramaya district, Eastern Ethiopia. BMC pediatrics 2015, 15(1):1-7.

---

## [Decision Letter · Decision Letter 3]

1 Feb 2023

PONE-D-22-10889R3Impact of fear of COVID-19 infection on home delivery among antenatal care booked women in Northeast EthiopiaPLOS ONE

Dear Dr. Asmamaw,

Thank you for submitting your manuscript to PLOS ONE. After careful consideration, we feel that it has merit but does not fully meet PLOS ONE’s publication criteria as it currently stands. Therefore, we invite you to submit a revised version of the manuscript that addresses the points raised during the review process.

We look forward to receiving your revised manuscript.

Kind regards,

Pracheth Raghuveer, MD, DNB

Academic Editor

PLOS ONE

Journal Requirements:

Reviewers' comments:

Reviewer's Responses to Questions

**Comments to the Author**

1. If the authors have adequately addressed your comments raised in a previous round of review and you feel that this manuscript is now acceptable for publication, you may indicate that here to bypass the “Comments to the Author” section, enter your conflict of interest statement in the “Confidential to Editor” section, and submit your "Accept" recommendation.

Reviewer #1: All comments have been addressed

Reviewer #3: All comments have been addressed

2. Is the manuscript technically sound, and do the data support the conclusions?

Reviewer #1: Yes

Reviewer #3: Partly

3. Has the statistical analysis been performed appropriately and rigorously? 

Reviewer #1: Yes

Reviewer #3: Yes

4. Have the authors made all data underlying the findings in their manuscript fully available?

Reviewer #1: Yes

Reviewer #3: No

5. Is the manuscript presented in an intelligible fashion and written in standard English?

Reviewer #1: Yes

Reviewer #3: No

6. Review Comments to the Author

Reviewer #1: (No Response)

Reviewer #3: Review Report

Title of the study: Impact of fear of COVID-19 infection on home delivery among antenatal care booked women in Northeast Ethiopia.

Objective: To assess the impact of fear of COVID-19 infection on home delivery among antenatal booked women in Northeast Ethiopia.

Revised as per comment: More than 90% have been addressed. However, still some of them needs brief inclusion into the revised paper. E.g., sentence that described “...women was had given birth” in line 39 and 40 needs meticulous edition.

Review Comments

1. We have report that states institutional service delivery was hampered as a result of fear of health institution related COVID-19 infection. Hence, what does “Impact of fear of COVID-19 infection on home delivery among antenatal care booked women” mean? Therefore, you are expected to reason out more the component of the titles and the title still needs meticulous revision ? If it is a must to continue with this type of title / I didn’t recommend/ it can be reframed as “unmarried women have high home delivery during the era of COVID-19 in Northeast Ethiopia”. The study area was wide. Hence, should be in some district or zone.

2. By the way the title and the dependent variable are inconsistent.

3. The key words are not complete.

4. In the background of the abstract, you have stated as “despite the importance of giving birth at health facilities, in Ethiopia, nearly half of the mothers who were booked for antenatal care gave birth at home”. Therefore, what was the evidence? Is that before or after the introduction of COVID-19? Hence, if they already half of them give birth at home before the pandemic? What is new then after the pandemics?

5. Why they are not included in women’s development army since it is comprehensive package of care for women?

6. Your objective, methods, results, discussions and the conclusions are not well consistent.

7. What is the philosophical stance of the reasearcher?

8. Many grammatical and language errors E.g., Stata to mean “STATA”.

9. Still the data quality measures need detail emphasis?

10. What are the measures taken to reduce recall bias?

11. Did the case to variable ratio fit? How did you control confounders and multi-collinearity? Why you presented mean?

12. Avoid repetitions. E.g., ‘declarations”.

13. The contents, statistics and language needs English language expert before the next submission.

Decision: Major Revision with extensive edition.

7. PLOS authors have the option to publish the peer review history of their article (what does this mean?). If published, this will include your full peer review and any attached files.

Reviewer #1: **Yes: **Saida Abrar

Reviewer #3: No

---

## [Author Response · Author response to Decision Letter 3]

4 Feb 2023

Authors’ response to the reviewer’s comments

Title: Unmarried women have high home delivery during the era of COVID-19 in Gidan district Northeast Ethiopia

Authors 

Desale Bihonegn Asmamaw (desalebihonegn1988@gmail.com)

Yohannes Ayanaw Habitu ((yohaneshabitu@gmail.com)

Eskedar Getie Mekonnen (eskedargetie18@gmail.com)

Wubshet Debebe Negash (wubshetdn@gmail.com)

Date: 04/022023

Authors’ Point-by-point response 

Dear reviewer, we would like to extend our deepest appreciation for devoting your time to reviewing our manuscript entitled “Unmarried women have high home delivery during the era of COVID-19 in Gidan district Northeast Ethiopia". There has been a revision of this manuscript (Abstract, introduction, methods, results, discussion, and conclusions). The whole structure of the manuscript has been revised. We hope now the manuscript is clear and more acceptable than its previous version. We have tried to present the paper in the proper manner according to your comment on what to suppose to do so. For this, here we have given our responses to each of the concerns you raised. Again, we would like to remind our strongest gratitude for your effort in the improvement of this manuscript, and all the points were addressed in the point-by-point response.

Reviewer comments

Reviewer comments: Revised as per comment: More than 90% have been addressed. However, still some of them needs brief inclusion into the revised paper. E.g., sentence that described “...women was had given birth” in line 39 and 40 needs meticulous edition.

Authors response: Thank you for your comments, this has been corrected accordingly. Kindly see line40-42.

Reviewer comments: We have report that states institutional service delivery was hampered as a result of fear of health institution related COVID-19 infection. Hence, what does “Impact of fear of COVID-19 infection on home delivery among antenatal care booked women” mean? Therefore, you are expected to reason out more the component of the titles and the title still needs meticulous revision ? If it is a must to continue with this type of title / I didn’t recommend/ it can be reframed as “unmarried women have high home delivery during the era of COVID-19 in Northeast Ethiopia”. The study area was wide. Hence, should be in some district or zone.

Authors response: Dear reviewer, thank you for your comment. The title was reframed as you suggested. Kindly see the updated version of the manuscript.

Reviewer comments: By the way the title and the dependent variable are inconsistent

Authors’ response: Dear reviewer, thank you for your observations. We corrected the errors. And the title and the dependent variable are consistent now. Kindly see the updated version of the manuscript.

Reviewer comments: The keywords are not complete.

Authors response: Thank you for your suggestion and observations. This has been addressed. Kindly see the updated version of the keywords.

Reviewer comments: In the background of the abstract, you have stated as “despite the importance of giving birth at health facilities, in Ethiopia, nearly half of the mothers who were booked for antenatal care gave birth at home”. Therefore, what was the evidence? Is that before or after the introduction of COVID-19? Hence, if they already half of them give birth at home before the pandemic? What is new then after the pandemics?

Authors’ response: Dear reviewer, thank you for your comments. In Ethiopia, nearly half of the mothers who were booked for antenatal care gave birth at home. This was reported by Ethiopia Demographic health survey (EDHS) of 2016. After the pandemic of COVID-19, the utilization of maternal health care like institutional delivery was more compromised [1]. This has been addressed in the updated version of the manuscript. Kindly see the updated version of the manuscript.

Reviewer comments: why they are not included in women’s development army since it is comprehensive package of care for women?

Authors’ response: Dear reviewer, thank you for your point of view. Yes, women’s development army is a comprehensive package of care for women. However, all women are not actively participated in the women’s development army. Only some of the women are actively involved in the package [2, 3]. Kindly consider these issues.

Reviewer comments: Your objective, methods, results, discussions and the conclusions are not well consistent.

Authors response: Thank you for your comments. This has been corrected accordingly. Kindly see the updated version of the manuscript.

Reviewer comments: What is the philosophical stance of the researchers?

Authors response: Thank you for your comments. Maternal morbidity and mortality are a global health challenge, and developing countries contribute to most maternal deaths. This can be reduced by improving maternal health care services utilizations like institutional service delivery. Previous scholars have stated that home delivery can be reduced by strengthening antenatal care (ANC) service utilization. But there are some studies reporting that ANC-booked women are still giving birth at home. And currently, places of delivery are also affected by the COVID-19 pandemic. Therefore, the aim of this study was to determine the prevalence and associated factors of home delivery among women who had antenatal care booked in their last pregnancy during the era of COVID-19.

Reviewer comments: Many grammatical and language errors E.g., Stata to mean “STATA”.

Authors response: Thank you for your observations. This has been corrected accordingly. Kindly see the updated version of the manuscript.

Reviewer comment: Still the data quality measures need detail emphasis?

Authors response: Thank you for your suggestions and comments. This has been addressed. Kindly see line 146-152.

Reviewer comments: What are the measures taken to reduce recall bias?

Authors response: Thank you for your comments. To reduce recall bias, some possible strategies were used, such as providing training for data collectors, employing pretests, and using standardized tools.

Reviewer comment: Did the case to variable ratio fit? How did you control confounders and multi-collinearity? Why you presented mean?

Authors response: Thank you for your comments. Prior to identifying the significant factors, multicollinearity was tested using the variance inflation factor (VIF), and we got a VIF of less than five for each independent variable with a mean VIF of 1.56, indicating there was no significant multicollinearity between independent variables. We used mean because, before selecting a summary measure, we checked for normality tests such as kurtosis and skewness were employed to see the normal distribution of the variables and to identify which summary measures were appropriate to use. We got a symmetric distribution, therefore we used mean. Kindly consider these issues.

Reviewer comment: Avoid repetitions. E.g., ‘declarations”.

Authors response: Thank you for your observations. The repetitions were avoided.

Reviewer comments: The contents, statistics and language needs English language expert before the next submission.

Authors response: Dear reviewer, thank you for your comments. An academic English language experts exhaustively proofread and edited the document. Kindly see the updated version of the manuscript.

References

1. Abdisa DK, Jaleta DD, Feyisa JW, Kitila KM, Berhanu RD: Access to maternal health services during COVID-19 pandemic, re-examining the three delays among pregnant women in Ilubabor zone, southwest Ethiopia: A cross-sectional study. Plos one 2022, 17(5):e0268196.

2. Wondimu MS, Woldesemayat EM: Determinants of home delivery among women in rural pastoralist community of hamar district, southern Ethiopia: a case–control study. Risk Management and Healthcare Policy 2020:2159-2167.

3. Zegeye EA, Reshad A, Bekele EA, Aurgessa B, Gella Z: The state of health technology assessment in the Ethiopian health sector: learning from recent policy initiatives. Value in Health Regional Issues 2018, 16:61-65.

Thank you

---

## [Decision Letter · Decision Letter 4]

18 Apr 2023

PONE-D-22-10889R4Unmarried women have high home delivery during the era of COVID-19 in Gidan district Northeast EthiopiaPLOS ONE

Dear Dr. Asmamaw,

Thank you for submitting your manuscript to PLOS ONE. After careful consideration, we feel that it has merit but does not fully meet PLOS ONE’s publication criteria as it currently stands. Therefore, we invite you to submit a revised version of the manuscript that addresses the points raised during the review process.

We look forward to receiving your revised manuscript.

Kind regards,

Pracheth Raghuveer, MD, DNB

Academic Editor

PLOS ONE

Journal Requirements:

Reviewers' comments:

Reviewer's Responses to Questions

**Comments to the Author**

1. If the authors have adequately addressed your comments raised in a previous round of review and you feel that this manuscript is now acceptable for publication, you may indicate that here to bypass the “Comments to the Author” section, enter your conflict of interest statement in the “Confidential to Editor” section, and submit your "Accept" recommendation.

Reviewer #3: All comments have been addressed

2. Is the manuscript technically sound, and do the data support the conclusions?

Reviewer #3: Partly

3. Has the statistical analysis been performed appropriately and rigorously? 

Reviewer #3: Yes

4. Have the authors made all data underlying the findings in their manuscript fully available?

Reviewer #3: Yes

5. Is the manuscript presented in an intelligible fashion and written in standard English?

Reviewer #3: Yes

6. Review Comments to the Author

Reviewer #3: Review Report

Title: Unmarried women have high home delivery during the era of COVID-19 in Gidan district Northeast Ethiopia.

Manuscript Number: PONE-D-22-10889R4.

Review Version: IV

Review Comments

General Comments

Which have high percentage E.g., unmarried women home delivery, rural residents home delivery, husband education with home delivery and not involving in women development army and home delivery or fear of COVID 19 and home delivery i.e. Why you prefer unmarried women over others? Taking the culture of the community and the occurrence of conflict in the area how do you say that most of them were unmarried since it is uncommon for the women to be pregnant to these much number? If that is so is that wanted pregnancy?

I prefer the title to be “ANC Booked Unmarried women have high Home delivery during the era of COVID 19 Pandemic in Gidan District, Ethiopia”

B. Specific Comments

1. Background: It well refined but still needs strengthening and shortening to the main objective of the study.

2. Methods

• The study period was not well written i.e., “March 30 and April 29, 2021”

• Have you conducted training of the data collectors to maintain quality assurance?

• Are there supervisors for the data collectors? How one PI supervisors over 760 sample?

• Who translated the two forms of the questionnaire?

• In the analysis have you calculated the case to variable ratio? If so, what was the result?

3. On the Result section

Brief, simple, logical and coherent way of writing the result.

Would you cross tab those associated factors with the home delivery.

The tables are not self-explanatory E.g. Didn’t describe the district.

4. Discussion and Conclusion

The discussion needs more refinement

There should be theoretical, practical, logical and frame work related explanations for those findings.

The conclusion should be drawn from the objective and implies the main issue of the study.

Where is whether there is funding or not.

5. Use of words: E.g., ‘North east Ethiopia’ should be Northeastern Ethiopia.

: Capital Letters E.g., kilometers should be Kilometers.

: Slided message: E.g., “who were booked for ANC”

: Some of the figures lack percentage after them E.g., “from 16 to 33%.”

: Inconsistencies E.g., Use of institutional delivery and health facility delivery and giving birth at health facilities and health institution delivery.

: Sentence flow gaps E.g., COVID-19 have disastrous effect.

6. Statistics

Still careful review on the fitness of the model, analysis and interpretations is needed.

Add cross tabulations

7. Language and Grammar

o Needs careful review

7. PLOS authors have the option to publish the peer review history of their article (what does this mean?). If published, this will include your full peer review and any attached files.

Reviewer #3: No

---

## [Author Response · Author response to Decision Letter 4]

25 Apr 2023

Authors’ Point-by-point response 

Dear reviewer, we would like to extend our deepest appreciation for devoting your time to reviewing our manuscript entitled "Antenatal care booked rural residence women have high home delivery during the era of COVID-19 pandemic in Gidan District, Ethiopia". The whole structure of the manuscript has been revised. We hope the manuscript is now clearer and more acceptable than its previous version. We have tried to present the paper in the proper manner according to your comment on how to do so. For this reason, we have given our responses to each of the concerns you raised.

Reviewer comments

Reviewer comments: Which have high percentage E.g., unmarried women home delivery, rural residents home delivery, husband education with home delivery and not involving in women development army and home delivery or fear of COVID 19 and home delivery i.e. Why you prefer unmarried women over others? Taking the culture of the community and the occurrence of conflict in the area how do you say that most of them were unmarried since it is uncommon for the women to be pregnant to these much number? If that is so is that wanted pregnancy?

Authors’ response: Dear reviewer, thank you for your concerns and suggestions. We appreciate your point of view; yes, you are right, and we modified the topic accordingly. We prefer rural residence women based on consideration of the odds ratio, the culture of the community, the occurrence of conflict in the area, and the accessibility of the services. Kindly see the updated version of the manuscript.

Reviewer comments: I prefer the title to be “ANC Booked Unmarried women have high Home delivery during the era of COVID 19 Pandemic in Gidan District, Ethiopia

Authors response: Dear reviewer, thank you for your suggestions. The title was reframed as accordingly. Kindly see the updated version of the manuscript.

Reviewer comments: Background: It is well refined but still needs strengthening and shortening to the main objective of the study

Authors’ response: Dear reviewer, thank you for your observations and comments. This has been addressed in detail. Kindly see the updated version of the manuscript.

Reviewer comments: The study period was not well written i.e., “March 30 and April 29, 2021”

Authors response: Thank you for your suggestion and observations. This has been corrected, kindly see the update version of it.

Reviewer comments: Have you conducted training of the data collectors to maintain quality assurance?

Authors’ response: Yes, the data collectors and supervisors were trained for two days, focusing on how to ask and fill out the questionnaires, the selection criteria for women, and how to approach the respondents. Kindly see page 9, line 144-147.

Reviewer comments: Are there supervisors for the data collectors? How one PI supervisors over 760 sample?

Authors response: There are eight BSc midwifery or nursing data collectors and two supervisors in the same field with experience in research and fieldwork coordination who participated in the data collection process. The whole data collection process was closely supervised by the principal investigator (PI). Kindly see page 9, line 160-163.

Reviewer comments: Who translated the two forms of the questionnaire?

Authors response: Thank you for your concerns. The two forms of the questionnaires were translated by language experts who have previous experience with this.

Reviewer comments: In the analysis have you calculated the case to the variable ratio? If so, what was the result?

Authors’ response: Yes, we calculated the case-to-variable ratio, the case-to-variable ratio showed 12.8 to 1, which indicates a ratio above the required 5 to 1. Kindly see the updated version of the manuscript.

Reviewer comments: The tables are not self-explanatory E.g. Didn’t describe the district.

Authors response: Thank you for your comments, this has been addressed, kindly see the updated tables.

Reviewer comments: The discussion needs more refinement, There should be theoretical, practical, logical and frame work related explanations for those findings.

Authors response: Dear reviewer, thank you for suggestions and comments, this has been addressed; kindly see the updated version of the manuscript.

Reviewer comments: Would you cross tab those associated factors with the home delivery.

Authors response: Dear reviewer, thank you for your concerns, this has been already done, kindly see table 3.

Reviewer comments: The conclusion should be drawn from the objective and implies the main issue of the study.

Authors response: Dear reviewer, thank you for your comments and suggestions, this has been addressed. 

Reviewer comments: Where is whether there is funding or not.

Authors response: This is available in the declaration section, kindly see page 17, line 313-314.

Reviewer comments: Use of words: E.g., ‘North east Ethiopia’ should be Northeastern Ethiopia

Capital Letters E.g., kilometers should be Kilometers

Slided message: E.g., "who were booked for ANC"

Some of the figures lack percentage after them E.g., "from 16 to 33%."

Inconsistencies E.g., Use of institutional delivery and health facility delivery and giving birth at health facilities and health institution delivery.

Sentence flow gaps E.g., COVID-19 have disastrous effect.

Authors response: Thank you for your observations and suggestions, these have been corrected and addressed, kindly see the updated version of the manuscript. 

Reviewer comments: Still careful review on the fitness of the model, analysis and interpretations is needed.

Authors response: Thank you for your comments, we were done the Hosmer-Lemeshow test, the model was adequate with the p-value of 0.26. Kindly see the updated version of the manuscript.

Reviewer comments: Language and Grammar

Authors response: Thank you for the suggestion, the entire article has been read and corrected by a native English speaker. Kindly see the updated version of the manuscript.

Thank you

---

## [Decision Letter · Decision Letter 5]

24 Jul 2023

PONE-D-22-10889R5Antenatal care booked rural residence women have high home delivery during the era of COVID-19 pandemic in Gidan District, EthiopiaPLOS ONE

Dear Dr. Asmamaw, 

Thank you for submitting your manuscript to PLOS ONE. After careful consideration, we feel that it has merit but does not fully meet PLOS ONE’s publication criteria as it currently stands. Therefore, we invite you to submit a revised version of the manuscript that addresses the points raised during the review process.

We look forward to receiving your revised manuscript.

Kind regards,

Pracheth Raghuveer, MD, DNB

Academic Editor

PLOS ONE

Journal Requirements:

Reviewers' comments:

Reviewer's Responses to Questions

**Comments to the Author**

1. If the authors have adequately addressed your comments raised in a previous round of review and you feel that this manuscript is now acceptable for publication, you may indicate that here to bypass the “Comments to the Author” section, enter your conflict of interest statement in the “Confidential to Editor” section, and submit your "Accept" recommendation.

Reviewer #4: (No Response)

2. Is the manuscript technically sound, and do the data support the conclusions?

Reviewer #4: Yes

3. Has the statistical analysis been performed appropriately and rigorously? 

Reviewer #4: Yes

4. Have the authors made all data underlying the findings in their manuscript fully available?

Reviewer #4: Yes

5. Is the manuscript presented in an intelligible fashion and written in standard English?

Reviewer #4: Yes

6. Review Comments to the Author

Reviewer #4: 1, ...Despite the importance

23 of giving birth at a health institution, in Ethiopia, according to the Ethiopian Demographic

24 Health Survey report, nearly half of the ANC-booked mothers gave birth at home

do not you think that the finding of EDHS data report of home birth is higher than the finding of this study? therefore, please omit the term high from the title of this research , for even it is below the national report, EDHS!

2, I don think that the sampling procedure is multi stage sampling because no more than one stage was gone to gate sampling unit for this study, u started from woreda then no more step to reach sampling unit simple u can select study unit SRS being at woreda level or kebeles due to the list already known and sharing the sample size proportionally for srata u randomly selected. there fore modify to simple random sampling.

3, repeated statements are there eg model adequacy

7. PLOS authors have the option to publish the peer review history of their article (what does this mean?). If published, this will include your full peer review and any attached files.

Reviewer #4: **Yes: **Jira Wakoya Feyisa

---

## [Author Response · Author response to Decision Letter 5]

28 Oct 2023

Authors’ response to the reviewer’s comments

Title: Antenatal care booked rural residence women have home delivery during the era of COVID-19 Pandemic in Gidan District, Ethiopia

Authors 

Desale Bihonegn Asmamaw (desalebihonegn1988@gmail.com)

Yohannes Ayanaw Habitu ((yohaneshabitu@gmail.com)

Eskedar Getie Mekonnen (eskedargetie18@gmail.com)

Wubshet Debebe Negash (wubshetdn@gmail.com)

Date: 22/04/022023

Authors’ Point-by-point response 

Dear reviewer, we would like to extend our deepest appreciation for devoting your time to reviewing our manuscript entitled "Antenatal care booked rural residence women have home delivery during the era of COVID-19 pandemic in Gidan District, Ethiopia". The whole structure of the manuscript has been revised. We hope the manuscript is now clearer and more acceptable than its previous version. We have tried to present the paper in the proper manner according to your comment on how to do so. For this reason, we have given our responses to each of the concerns you raised.

Reviewer comments

Reviewer comments: , ...Despite the importance of giving birth at a health institution, in Ethiopia, according to the Ethiopian Demographic Health Survey report, nearly half of the ANC-booked mothers gave birth at home do not you think that the finding of EDHS data report of home birth is higher than the finding of this study? therefore, please omit the term high from the title of this research , for even it is below the national report, EDHS!

Authors response: Dear reviewer, thank you for your concerns, we have been addressed this issue according to your comments, kindly see the updated version of the manuscript.

Reviewer comments: I don think that the sampling procedure is multi stage sampling because no more than one stage was gone to gate sampling unit for this study, u started from woreda then no more step to reach sampling unit simple u can select study unit SRS being at woreda level or kebeles due to the list already known and sharing the sample size proportionally for srata u randomly selected. there fore modify to simple random sampling.

Authors response: Dear reviewer, thank you for your observation and suggestion, we have been modified according to your comment and suggestion, kindly see the updated version of the manuscript.

Reviewer comments: repeated statements are there eg model adequacy

Authors’ response: Dear reviewer, thank you for your observations, the errors were corrected, kindly see the updated versions of the manuscript.

Thank you

---

## [Editor Report · Decision Letter 6]

20 Nov 2023

Antenatal care booked rural residence women have home delivery during the era of COVID-19 pandemic in Gidan District, Ethiopia

PONE-D-22-10889R6

Dear Dr. Desale

We’re pleased to inform you that your manuscript has been judged scientifically suitable for publication and will be formally accepted for publication once it meets all outstanding technical requirements.

Kind regards,

Pracheth Raghuveer, MD, DNB

Academic Editor

PLOS ONE
---

## [Editor Report · Acceptance letter]

27 Nov 2023

PONE-D-22-10889R6 

Antenatal care booked rural residence women have home delivery during the era of COVID-19 pandemic in Gidan District, Ethiopia 

Dear Dr. Asmamaw:

I'm pleased to inform you that your manuscript has been deemed suitable for publication in PLOS ONE. Congratulations! Your manuscript is now with our production department. 

Kind regards, 

on behalf of

Dr. Pracheth Raghuveer 

Academic Editor

PLOS ONE